# Towards Robust Computation of Cardiothoracic Ratio from Chest X-ray

**Matilde Bodritti**[1,2]*                    MATILDEBODRITTI98@GMAIL.COM
[1] *Agfa Radiology Solutions, Agfa NV, Belgium*
[2] *Ghent University, Ghent, Belgium*

**Adriyana Danudibroto**[1]                    ADRIYANA.DANUDIBROTO@AGFA.COM

**Jan Aelterman**[3,4,5]                    JAN.AELTERMAN@UGENT.BE
[3] *Ghent University Centre for X-Ray Tomography (UGCT), Proeftuinstraat 86/N12, Ghent, Belgium*
[4] *IPI-TELIN-IMEC, Ghent University, Ghent, Belgium*
[5] *Radiation Physics Research Group, Department of Physics and Astronomy, Ghent University, Proeftuinstraat 86, Ghent, Belgium*

## Abstract

The cardiothoracic ratio (CTR) plays an important role in early detection of cardiac enlargement related diseases in chest X-ray (CXR) examinations. Since its measurement would be time-consuming, its evaluation in clinical practice is done by a visual assessment: it is highly subjective and its robustness is undermined by some acquisition issues such as lung clipping or patient orientation variation. No work addressing the problem of clipped lungs in the CTR estimation has been found in the literature. For these reasons, aiming for a robust method, we firstly proposed a segmentation-based approach for automatic measurement of the CTR (based only on the lung segmentation mask) able to handle clipped anatomy cases. Secondly, the proposed method was validated on a large dataset allowing us to corroborate earlier research results with manual CTR computation in which the mean CTR increases with the age of the patients and there is a noticeable difference between men and women's CTR. Lastly, a new rotational invariant metric was proposed, showing it to be more robust to different patient orientations.

**Keywords:** cardiothoracic ratio, chest anatomy segmentation, chest X-ray

## 1. Introduction

Chest X-ray (CXR) is the most commonly performed diagnostic X-ray examination. However, its low diagnostic sensitivity (when compared to cross-sectional techniques) needs to be counterbalanced by an accurate and time-consuming radiologist interpretation. This can be helped by computer-aided technologies which instead of outputting directly the disease inferred from the CXR, they can output some measurements (as objective as possible) that will help the clinician to formulate the diagnosis. An example of objective measurement is the cardiothoracic ratio (CTR): a screening tool to evaluate the size of the heart's silhouette and thus the presence of cardiomegaly from CXR. The theoretical definition of the CTR involves calculating the ratio between the maximum horizontal heart diameter (Dheart) and the maximum horizontal thoracic diameter (Dthorax). In the literature, almost all approaches to automatically extract the CTR have the underlying assumption that the images

---

* The work was conducted during an internship and M.Sc. training at 1 and 2.

are taken from correct acquisitions. However, if part of the lung area is outside image's field of view, the measurement of Dthorax can be affected: this is one issue to take into account for a robust estimation. For this reason, we choose to explore the computation of CTR in case of clipped anatomy. Even if multiple CXR datasets are publicly available, only few of them has lung and heart masks annotations: we choose to extract heart shape information from only the contour of the lungs, unlike most of the works in literature that rely on both heart and lungs mask segmentations (Gupte et al., 2021).

## 2. Materials and Methods

Starting with the CXR image, the lung segmentation mask is extracted. From the lung segmentation mask, both Dheart and Dthorax are extracted and the CTR is calculated. Dheart is defined as the maximum horizontal distance between the two lungs, above the vertex of the cardiophrenic angle, as shown in Figure 1.A. For the segmentation task we modified the U-Net with variational autoencoder by Selvan et. al (Selvan et al., 2020). The modification allows an output with a field of view 128 pixels wider on each side than the input image to handle cases in which anatomy is partially clipped out of the image. The implementation details and dataset used can be found on the GitHub repository.

The proposed method was then applied to a large dataset to be validated. These types of population studies are usually difficult to carry out on a large scale, because of the need of clear and structured radiologists annotations for each CXR. The automatic calculation of CTR can make this process faster and easily accessible. A subset of the CheXpert dataset (Irvin et al., 2019) was selected, resulting in 25,369 CXRs with theoretically normal values of CTRs, comprised of 34% female and 66% male, from 18 to 90-year-old.

Since the evaluation of the CTR is used in everyday clinical practice, we also wanted to evaluate the robustness of this metric. The previously described CTR calculation method is highly dependent on the orientation of the performed acquisition and for this reason, a different metric, strongly related to CTR has been proposed: the rotational invariant CTR (RI_CTR). This method also involved the estimation of heart's contour and the details of the implementation can be found on the GitHub repository. Dheart is now defined as the diameter of the maximum circle inscribed in the heart masks, while Dthorax is the maximum horizontal width of the rotated lungs, as shown on Figure 1.B. The orientation of the lung mask is derived by finding the major axis of the mask. Then, it is oriented to 0 degrees to obtain consistent orientation. The performance of this method have been tested on lung and heart masks from clipped and non-clipped test sets in terms of absolute error, root mean square error and correlation coefficient.

## 3. Results and Discussion

As a baseline, the CTR calculated using the segmentation model by Selvan et. al (Selvan et al., 2020), resulted in an absolute error of $0.074 \pm 0.090$ on a test set with clipped lungs. The proposed method for CTR estimation reported an absolute error of $0.058 \pm 0.057$ on the same test set. The performance of the proposed method was in the same order

of magnitude compared to other state-of-the-art method that computes CTRs from lungs segmentations (Dallal et al., 2017).

The CTR values obtained by the application of the method on the CheXpert subset are shown on Figure 1.C. The variation of mean CTR with age and gender reflects the observations of previous studies based on manually annotated CTRs (Brakohiapa et al., 2021). They reported a significant difference in the overall CTR between men and women, with a slightly higher mean CTR and a higher increase in mean CTR values for women as age increases. Both trends are reflected in our results. This suggests that the proposed method could be suitable for such population studies.

Moreover, the proposed RI_CTR shows a much higher correlation coefficient (CC) with the RI_CTR calculated from ground truth segmentation masks when compared to the CTR, and it also shows lower absolute error (AE) and root mean square error (RMSE). For the CTR method we reported a CC of 0.558, an AE of $0.062 \pm 0.059$ and a RMSE of 0.086, while for the RI_CTR method we obtained a CC of 0.754, an AE of $0.024 \pm 0.021$ and a RMSE of 0.031. The lower errors identified for the RI_CTR method compared to the CTR method may indicate that it is indeed much more robust as a metric, yet future research is needed to establish RI_CTR as an alternative clinical metric.

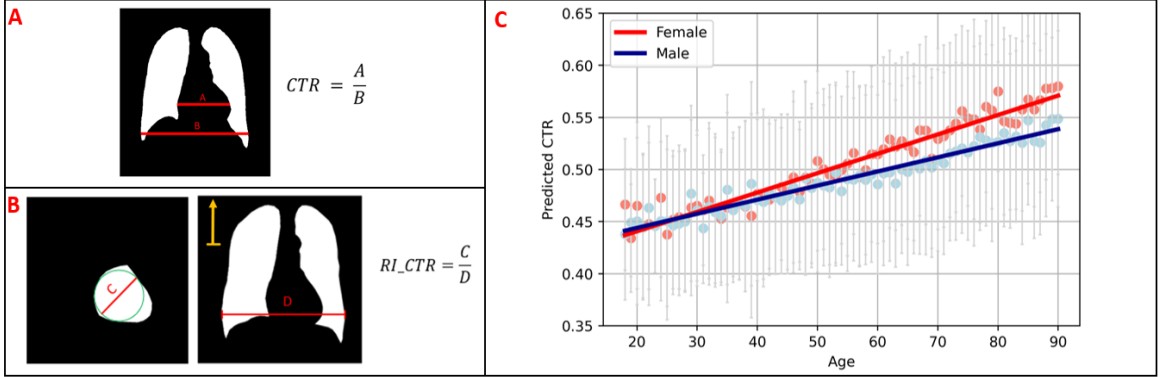

Figure 1: **A)** Illustration of CTR calculation from lung segmentation mask. **B)** Illustration of RI_CTR calculation from lung and heart segmentation masks. The yellow arrow represents the major axis of the lung. **C)** Predicted CTR as a function of patient age on men and women's CXRs from CheXpert subset.

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
