# OpenReview forum: "Towards Robust Computation of Cardiothoracic Ratio from Chest X-Ray"
_MIDL.io/2023/Short_Paper_Track — MIDL 2023 Short paper track Poster_

### Official Review · Reviewer_yNQ6 · 2023-04-15
**little technical novelty, key details missing, conclusions not supported by results**

**Rating:** 4
**Confidence:** 4

**Review:**

No technical novelty

Methodological problem: Correlation coefficient being computed between CTR and ground truth RI_CTR < correlation coefficient between predicted RI_CTR and ground truth RI_CTR: this does not prove that RI_CTR is more robust -- indeed it is expected since CTR and RI_CTR are computed in different ways.

A lot of vital details are missing:
1. What is the test set on which the effect of clipped lungs is evaluated? the paper reports some improvement in terms of the absolute error compared to baseline method
2. How were the ground truth segmentations obtained (for computing CC for CTR and RI_CTR)?

---

### Official Review · Reviewer_9Dai · 2023-04-19
**Good article on robust CTR estimation from x-ray**

**Rating:** 7
**Confidence:** 4

**Review:**

The article presents a technique for measuring the cardiothoracic ratio (CTR) from chest X-ray images.
The authors specifically address two difficulties:
- lungs can be clipped (outside of field of view)
- patient can be mis-orientated during acquisition
The approach was validated against a large (N=25,000) public database

Some details are provided in the accompanying repository:
https://anonymous.4open.science/r/Robust_computation_CTR/README.md

Pros:
* The authors objectives are clear and the paper is easy to read
* Validation on a large dataset provides confidence in the method's performance and generalizability.

Cons:
* An analysis separating the benefits of correcting for clipping and mis-orientation would be useful

Ideas for related and future work:
* Could the complete implementation of the technique be open-sourced?
* Conduct further research on the clinical utility of the technique